# Vectors on the Move: How Climate Change Fuels the Spread of Arboviruses in Europe

**DOI:** 10.3390/microorganisms13092034

**Published:** 2025-08-30

**Authors:** Giulia Carbone, Giulia Boiardi, Claudia Infantino, Daniela Cunico, Susanna Esposito

**Affiliations:** Pediatric Clinic, Department of Medicine and Surgery, University of Parma, 43126 Parma, Italy; giulia.carbone@unipr.it (G.C.); boiardigiulia@gmail.com (G.B.); infantinoclaudia@gmail.com (C.I.); dani.cunico@gmail.com (D.C.)

**Keywords:** arboviruses, climate change, vector-borne diseases, emerging infectious diseases, mosquito-borne viruses

## Abstract

Climate change is increasingly recognized as a major driver of emerging infectious diseases, particularly vector-borne diseases (VBDs), which are expanding in range and intensity worldwide. Europe, traditionally considered low-risk for many arboviral infections, is now experiencing autochthonous transmission of pathogens such as dengue, chikungunya, Zika virus, West Nile virus, malaria, and leishmaniasis. Rising temperatures, altered precipitation patterns, and milder winters have facilitated the establishment and spread of competent vectors, including *Aedes*, *Anopheles*, *Phlebotomus*, and *Culex* species, in previously non-endemic areas. These climatic shifts not only impact vector survival and distribution but also influence vector competence and pathogen development, ultimately increasing transmission potential. This narrative review explores the complex relationship between climate change and VBDs, with a particular focus on pediatric populations. It highlights how children may experience distinct clinical manifestations and complications, and how current data on pediatric burden remain limited for several emerging infections. Through an analysis of existing literature and reported outbreaks in Europe, this review underscores the urgent need for enhanced surveillance, integrated vector control strategies, and climate-adapted public health policies. Finally, it outlines research priorities to better anticipate and mitigate future disease emergence in the context of global warming. Understanding and addressing this evolving risk is essential to safeguard public health and to protect vulnerable populations, particularly children, in a rapidly changing climate.

## 1. Introduction

Climate change has emerged as one of the greatest global health threats of the 21st century. The World Health Organization (WHO) recognizes it as a critical risk factor for human health, with the potential to exacerbate both communicable and non-communicable diseases [1]. Recent decades have witnessed an alarming acceleration of global warming: by 2020, average global temperatures had increased by approximately 2.14 °C compared to the pre-industrial baseline (1880–1900), with the ten hottest years on record all occurring since 2005 [2]. This rise is largely attributed to anthropogenic activities such as fossil fuel combustion and deforestation [3]. Without significant reductions in greenhouse gas emissions, global temperatures are projected to rise by an additional 1.5 °C to 2 °C by the end of the century [3].

The impact of climate change extends far beyond temperature increases. It includes shifts in precipitation patterns, prolonged droughts, more frequent and severe heatwaves, and increased intensity of extreme weather events [4]. These environmental alterations have profound implications for infectious disease ecology, particularly in relation to vector-borne diseases (VBDs). VBDs are especially sensitive to climate variability because the life cycles of vectors, pathogens, and hosts are all climate-dependent [5].

In Europe, these changes have already begun to shift the epidemiology of VBDs. Warming temperatures and altered rainfall have facilitated the northward expansion of vectors such as *Aedes* spp. mosquitoes and *Phlebotomus* spp. sandflies [6]. This has increased the risk of local transmission of arboviral diseases such as dengue, chikungunya, and Zika. At the same time, climate change is influencing the epidemiology of VBDs that are already autochthonous in Europe, including leishmaniasis (transmitted by sandflies) and West Nile virus (transmitted by *Culex* mosquitoes) [6]. The introduction of these diseases into immunologically naïve populations—combined with variable public health preparedness—poses a growing challenge for surveillance, diagnosis, and control efforts.

Although several studies have explored the association between climate change and infectious diseases, most have focused on global or adult populations. There is still a lack of comprehensive overviews specifically addressing the emergence of VBDs in Europe and their potential impact on pediatric populations, who may be particularly vulnerable due to immunologic, developmental, or social factors. This narrative review aims to address these unmet needs by summarizing current evidence on how climate change is influencing the transmission and distribution of vector-borne diseases in Europe.

## 2. Methods

This narrative review was conducted to synthesize the current evidence on the impact of climate change on the emergence, transmission, and geographic spread of VBDs in Europe, with particular attention to pediatric implications. An electronic search was performed using the PubMed database, covering publications from 1 January 2010, to 30 June 2025. The search strategy combined terms related to vectors (e.g., “vector”, “mosquito”, “tick”, “sandfly”), pathogens (e.g., “virus”, “bacteria”, “parasite”), specific diseases (e.g., “West Nile”, “malaria”, “leishmania”, “dengue”, “zika”, “chikungunya”), and climate-related terms (e.g., “climate change”, “global warming”, “temperature”, “precipitation”), along with epidemiological and clinical keywords such as “infectious disease”, “pediatric”, and “child”. Boolean operators “AND” and “OR” were used to broaden and refine the search.

Only peer-reviewed articles written in English were considered. After the initial search, the reference lists of selected papers and reviews were manually screened to identify additional relevant publications not captured in the database search. Eligible articles included original research, systematic reviews, meta-analyses, surveillance reports, and clinical studies that addressed the relationship between climate variables and vector-borne diseases, with a particular focus on the European context. Studies focusing solely on non-European regions, non-peer-reviewed sources, commentaries, and editorials were excluded. Although our search strategy also included ticks as vectors, the present review was deliberately limited to arboviruses and selected parasitic diseases (e.g., malaria and leishmaniasis). Tick-borne infections, such as Lyme disease and tick-borne encephalitis, were not addressed in detail because they warrant a separate, dedicated review given their distinct epidemiology, ecology, and clinical spectrum.

Two reviewers independently (CI and DC) screened titles and abstracts for relevance. Full texts of potentially eligible studies were then assessed to determine inclusion. In cases of disagreement, consensus was reached through discussion or consultation with a third reviewer (GC). From each included study, data were extracted regarding the type of vector and pathogen, geographic region, relevant climate variables, evidence of correlation between climate and disease patterns, and pediatric outcomes when available.

The selected evidence was analyzed qualitatively and organized by disease type and climate-related drivers such as temperature changes, rainfall variability, or altered seasonality. This approach allowed for an integrated synthesis of findings from multiple disciplines, including epidemiology, climatology, entomology, and pediatric infectious diseases, in order to better understand the evolving threat of vector-borne diseases in the context of climate change in Europe.

## 3. Transmission Dynamics of Vector-Borne Diseases (VBDs)

VBDs represent a substantial global public health threat. Their transmission dynamics are intricately dependent on environmental factors, particularly climate variables such as temperature, humidity, and precipitation. Arthropod vectors—primarily mosquitoes, but also ticks and sandflies—serve as the main transmitters of VBDs. As ectothermic (cold-blooded) organisms, these vectors are especially sensitive to fluctuations in environmental temperatures and weather conditions, which influence their development, behavior, survival, and competence to transmit pathogens [5]. Table 1 summarizes the impact of climate change on VBDs.

All three components involved in disease transmission—vector, pathogen, and host—are highly responsive to climate variability. Numerous studies have demonstrated strong correlations between local climatic shifts and the incidence or prevalence of VBDs in affected regions [5]. In Europe, climate change has led to observable alterations in temperature and precipitation regimes, including an increased frequency and intensity of heatwaves, longer and warmer summers, and more irregular rainfall patterns, all of which significantly affect the transmission dynamics of VBDs [6].

Elevated temperatures alter not only the geographic distribution of vectors but also their developmental cycles, feeding behavior, and interactions with hosts and pathogens. Some pathogens may be lost from vectors in changing climates, while others may emerge or expand due to the enhanced suitability of environmental conditions [7]. Transmission typically occurs after vectors ingest pathogens during a blood meal from an infected host. These pathogens then replicate or circulate within the vector until it becomes infectious [8]. In the case of arboviruses, this internal replication phase is termed the extrinsic incubation period (EIP), which is highly temperature-dependent [8].

Vector competence—the intrinsic ability of a vector to acquire, maintain, and transmit a pathogen—is influenced by both genetic and environmental factors, particularly climate [9]. Because insects lack thermoregulation, their physiological functions are tightly linked to ambient temperatures, which must reach specific thresholds for essential biochemical processes to occur [10].

Viral replication within vectors is also temperature-sensitive. It begins above a minimum thermal threshold and increases with rising temperatures, up to an upper limit beyond which replication efficiency declines [11]. Experimental studies have shown that the EIP tends to decrease with increasing temperatures, accelerating pathogen transmission—although this relationship may reverse at extreme heat levels [12,13]. Additionally, fluctuating temperatures can indirectly influence disease transmission by affecting vector longevity, reproductive capacity, and host-seeking behavior, as well as by promoting insecticide resistance [14,15].

As temperatures rise, the geographic range and seasonal activity of key vectors such as mosquitoes and ticks have expanded. This has facilitated the emergence or re-emergence of diseases, including dengue, chikungunya, and malaria in regions previously unsuitable for their transmission [16,17,18]. Warmer winters and earlier springs have extended the active periods of vectors, thereby lengthening the transmission season [19,20].

Increased temperatures can also boost reproductive rates of vectors, leading to greater vector densities and consequently higher transmission risk [17]. Moreover, changes in precipitation patterns can modify breeding habitats; for example, heavy rains or irregular rainfall can result in the formation of stagnant water pools, ideal for mosquito oviposition [18].

Climate-induced shifts in the distribution of both vectors and pathogens can introduce diseases into immunologically naïve populations, potentially resulting in large-scale outbreaks [17]. Studies have shown that temperature fluctuations influence the behavior of the dengue virus by enhancing mosquito flight range, extending daily activity periods, and shortening the EIP, which collectively increase transmission efficiency [21]. Similarly, the increasing incidence of malaria has been linked to climate-related changes in rainfall and temperature, particularly in regions where the disease had previously been controlled or eradicated [22].

Table 2 shows the emerging VBDs in Europe.

## 4. Clinical Manifestations of the Most Frequent Vector-Borne Diseases (VBDs)

### 4.1. Dengue

Dengue is a febrile illness caused by four antigenically distinct but genetically related flaviviruses (DENV-1–4) transmitted by Aedes aegypti and Aedes albopictus mosquitoes [23,24]. Infection with one serotype confers lifelong immunity to that type but only transient cross-protection against others, with antibody-dependent enhancement increasing the risk of severe disease upon secondary infection [25]. Globally, most cases occur in the Asia-Pacific region (≈70%), followed by Africa (16%) and the Americas (14%) [26]. While DENV is not endemic in Europe, the presence of competent vectors poses a persistent risk of local outbreaks. Autochthonous cases have been documented in France and Italy, alongside numerous imported cases, including 185 in Italy in 2019 and 450 in 2024 (425 imported, 25 local), with no reported deaths [26,27,28]. The global incidence of dengue has risen sharply due to urbanization, globalization, poor sanitation, and increased mobility. WHO estimates ≈390 million annual infections across 128 countries, making dengue the fastest-spreading mosquito-borne viral disease, with 5–6 billion people projected at risk by 2050 [29,30].

Climate change is a significant concern in dengue epidemiology. Higher temperatures accelerate the mosquito life cycle and reduce the EIP, enhancing transmission potential [24,31]. The bidirectional transmission cycle between viremic humans and competent mosquitoes is sensitive to human mobility, urban density, and seasonal changes. During the COVID-19 pandemic, reductions in mobility led to marked declines in dengue incidence, highlighting the influence of human behavior on transmission dynamics [32].

Key factors contributing to DENV spread include increased vector density in rainy seasons, shortened EIP at elevated ambient temperatures, higher density of susceptible hosts, and prolonged human viremia [33]. The risk of infection increases by 13% for each 1 °C rise in temperature above the baseline [30,34,35,36]. Vector dispersal through human travel and global trade also contributes to the virus’s reach.

Dengue transmission follows two patterns: epidemic (sporadic introductions of single viral strains) and hyperendemic (co-circulation of multiple serotypes). Epidemic dengue typically affects both children and adults during localized outbreaks, often originating in seaports before WWII [25,33]. Hyperendemic dengue, found in densely populated tropical regions with continuous vector presence, is associated with higher seroprevalence in adults and a shift in symptomatic cases to the pediatric population [37,38,39,40].

DENV is transmitted primarily by *Ae. aegypti*, although *Ae. albopictus* can also serve as a competent vector, especially in temperate regions. *Ae. albopictus* is more tolerant of cold climates but less anthropophilic and less efficient in DENV transmission compared to *Ae. aegypti* [25,33,41,42,43,44]. Both species are also competent vectors for Zika and chikungunya viruses, enabling co-circulating outbreaks [45].

Clinically, dengue presents a wide range of manifestations. Up to 80% of infections are asymptomatic, while symptomatic cases typically manifest 4–10 days after the mosquito bite. Classic symptoms include high fever, retro-orbital pain, myalgia, arthralgia, rash, and gastrointestinal symptoms. Severe cases may evolve into DHF/DSS, characterized by plasma leakage, hemorrhagic manifestations, and potentially fatal hypovolemic shock. Mortality is under 1% with appropriate hospital management [46,47].

In children, symptoms are often nonspecific. Younger children may present with irritability rather than the classic retro-orbital pain or arthralgia. Prommalikit et al. reported higher frequencies of hepatomegaly, diarrhea, convulsions, and rash in children compared to adults [48]. Malavige et al. observed greater fluid loss in pediatric patients, while Jayarajah et al. linked gastrointestinal symptoms and leukopenia to DHF in children [49,50]. Rocha et al. noted age-specific hematologic changes, with younger children having higher hemoglobin levels, while older children showed anemia during symptomatic phases [51].

There are no specific antivirals for dengue. Management focuses on supportive care, fluid balance, and antipyretics (paracetamol preferred). Recovery confers serotype-specific lifelong immunity, but reinfection with a different serotype increases the risk of severe disease [52].

Two dengue vaccines have been approved. CYD-TDV (Dengvaxia) is a live attenuated tetravalent vaccine recommended only for individuals with confirmed prior dengue infection, due to the risk of severe disease upon first infection post-vaccination in DENV-naïve individuals [53,54,55,56]. TAK-003 (Qdenga) is another tetravalent vaccine approved in multiple countries, including the EU (≥4 years of age), Brazil, Thailand, and Indonesia [56,57].

### 4.2. Chikungunya

Chikungunya virus (CHIKV) is an arthropod-borne RNA virus of the genus Alphavirus (family *Togaviridae*), first isolated in Tanzania in 1953 [58]. It is transmitted mainly by Aedes aegypti and *Ae. albopictus*, with non-human primates serving as reservoirs, while humans can act as amplifying hosts during outbreaks [59,60]. CHIKV is endemic in tropical and subtropical regions of Africa and Southeast Asia [61] and has diverged into two major lineages: West African, associated with enzootic cycles and limited outbreaks, and East/Central/Southern African (ECSA), which has spread widely and gave rise to the Asian genotype [62,63].

Historically transmitted by *Ae. aegypti*, CHIKV has expanded its range through viral adaptations—particularly the E1-A226V mutation—that enhanced replication in *Ae. albopictus*, a vector tolerant of temperate climates. This shift has facilitated outbreaks beyond the tropics, including in continental Europe [64,65,66].

The first autochthonous European outbreak occurred in 2007 in Ravenna, Italy, following the introduction of the virus by a viremic traveler returning from India [67]. Subsequent local transmission events were reported in southern France in 2010, where two children with no travel history became infected, implicating *Ae. albopictus* as the vector [68,69]. Climatic projections suggest that environmental suitability for *Ae. albopictus* in Europe will continue to expand due to ongoing climate change [70,71]. Further local outbreaks occurred in Italy’s Lazio and Calabria regions during the summer of 2017, while in the Americas, CHIKV spread rapidly after 2013, reaching the Caribbean, South America, and parts of the southern United States [72].

Fisher et al. highlighted the importance of understanding the EIP—the time from viral ingestion by the mosquito to transmission capability—as a key parameter in modeling transmission risk. While temperature-dependent EIP estimates for DENV are well-characterized and mapped across Europe, equivalent experimental data for CHIKV remain limited [69]. Although there is growing consensus on the sensitivity of VBDs to climate change, few predictive studies have modeled future spatiotemporal CHIKV dynamics under European climate change scenarios [69].

CHIKV infection typically manifests as an acute febrile illness with rash and severe polyarthralgia. Although the clinical picture can resemble dengue fever, recurrent and prolonged musculoskeletal symptoms—predominantly involving peripheral joints—are a hallmark of chikungunya and can persist for months or even years post-infection [61]. While most cases are self-limiting, increasingly severe clinical presentations have been observed in recent years, including neurological complications, fulminant hepatitis, and neonatal encephalopathy [61].

In pediatric populations, symptomatic CHIKV infections are mainly reported in children over two years of age [73]. The clinical phenotype commonly includes fever, rash, and arthralgia, with gastrointestinal and neurological symptoms also noted. Although children generally experience milder courses than adults, they may be more prone to prolonged joint symptoms or febrile seizures, both of which can adversely affect quality of life [74]. Furthermore, congenital and perinatal infections have been documented since 2005 in neonates born to viremic mothers. These cases have exhibited severe manifestations such as dehydration, cardiac anomalies, seizures, neurologic sequelae, and, in rare cases, mortality [73].

Children may be disproportionately affected by climate-driven changes in vector distribution due to their heightened vulnerability to severe disease forms and complications [74]. Therefore, pediatric surveillance is essential in understanding and mitigating the full clinical impact of CHIKV in the context of a changing environment.

At present, the most effective strategy for preventing local transmission in Europe is rigorous syndromic, clinical, and laboratory-based surveillance, especially targeting viremic individuals returning from endemic areas. In regions where *Ae. albopictus* is established, such measures are critical to avert autochthonous outbreaks and reduce vector-human transmission cycles [75].

### 4.3. Zika

Zika virus disease is a mosquito-borne illness caused by Zika virus (ZIKV), a positive-sense, single-stranded RNA virus belonging to the genus Flavivirus, family *Flaviviridae*, within the *Spondweni* serocomplex. It was first identified in 1947 in a rhesus monkey in the Zika forest of Uganda, later in *Ae. africanus* mosquitoes in 1948, and in humans in Nigeria in 1952 [74]. ZIKV is phylogenetically and antigenically related to other flaviviruses such as DENV, yellow fever virus, Japanese encephalitis virus, and West Nile virus. Two major phylogenetic lineages have been described: the African lineage and the Asian lineage, the latter of which has emerged in the Pacific and subsequently the Americas [76,77].

ZIKV distribution prior to 2007 was largely confined to equatorial Africa and parts of Southeast Asia. Its known presence was based primarily on serological evidence and virus isolation in mosquitoes and humans. The first documented outbreak outside these regions occurred on Yap Island, Federated States of Micronesia, in 2007 [77]. Between 2013 and 2015, the virus caused outbreaks across the Pacific Islands, most notably in French Polynesia. In 2015, it emerged in South America, leading to a large epidemic in Brazil, with an estimated 1.3 million infections and subsequent spread across the Americas [78,79,80,81,82]. In Italy, 12 imported cases of ZIKV infection were reported in 2016 [83]. No pediatric data are currently available for the Italian population.

Following the 2015–2017 epidemic, no major outbreaks have been reported. Predicting future outbreaks remains challenging due to factors including under-reporting of asymptomatic cases, cross-immunity from other flaviviruses, and incomplete understanding of the duration of post-infection immunity, which appears to be protective but may wane over time [84].

The primary vectors of ZIKV are *Aedes* mosquitoes, especially *Ae. aegypti*, which exhibit a strong anthropophilic behavior and are well adapted to urban habitats. These mosquitoes tolerate higher temperatures than *Anopheles* spp., implying that climate change may favor their wider distribution. Projections suggest that by 2080, over one billion more people—primarily in North America and Europe—could be exposed to Aedes-borne arboviruses, including ZIKV, due to rising global temperatures and urbanization [85].

Although *Ae. aegypti* is the principal vector, several other *Aedes* species, such as *Ae. albopictus*, *Ae. africanus*, *Ae. hensilli*, *Ae. polynesiensis*, *Ae. unilineatus*, and *Ae. vittatus*, have shown potential vector competence either experimentally or in the field [76,77]. Apart from vector-borne transmission, ZIKV can be transmitted vertically from mother to fetus via transplacental infection, intrapartum during delivery, through sexual contact, and via transfusion of infected blood products or organ transplantation [76,77,86].

After a bite from an infected mosquito, the EIP typically ranges from 3 to 12 days. Most ZIKV infections (approximately 80%) are asymptomatic. When symptomatic, the disease is typically self-limiting, lasting 2 to 7 days. Clinical manifestations are generally mild and include low-grade fever, maculopapular rash (often beginning on the face), conjunctival injection (non-purulent), arthralgia, myalgia, fatigue, and headache. Retro-orbital pain and mild gastrointestinal symptoms may also occur in some cases [87]. Given the overlapping symptomatology, differential diagnosis includes dengue, chikungunya, measles, rubella, parvovirus B19, enteroviruses, and malaria. Co-infection with DENV or CHIKV may also occur in endemic areas [87,88].

A significant concern regarding ZIKV is its teratogenic potential. During the Brazilian outbreak in 2016–2017, an increase in congenital abnormalities was observed, leading to the identification of congenital Zika syndrome (CZS). CZS includes severe microcephaly, craniofacial disproportion, brain calcifications, ventriculomegaly, cortical atrophy, congenital contractures (e.g., arthrogryposis), visual and auditory deficits, epilepsy, and dysphagia [89]. Although not all infants born to ZIKV-infected mothers develop abnormalities, some children without overt microcephaly at birth demonstrated cognitive delays, suggesting the potential for neurodevelopmental effects beyond structural defects [89].

Diagnosis of ZIKV infection relies on molecular methods such as reverse transcription-polymerase chain reaction (RT-PCR) to detect viral RNA in clinical specimens. Viremia is typically detectable in serum and saliva during the first 3–7 days after symptom onset and in urine for up to 2–3 weeks. Viral RNA can also be detected in semen for up to 62 days or more, underscoring the potential for sexual transmission [90,91,92].

Preventive measures primarily aim to reduce vector exposure. Personal protection includes the use of repellents, long-sleeved clothing, insecticide-treated nets, and avoidance of mosquito-infested areas during peak biting times, which for *Aedes* species typically occur during daylight hours and twilight [93,94]. Vector control strategies emphasize eliminating mosquito breeding sites, especially artificial containers with stagnant water near human dwellings [44,95,96].

People returning from Zika-endemic regions are advised to continue mosquito bite precautions for at least three weeks after travel, to minimize the risk of local transmission if they become viremic. During the first week of illness, infected individuals should strictly avoid further mosquito exposure to prevent the onward transmission of the virus to local vectors [93,95].

### 4.4. Malaria

Malaria is caused by protozoa of the genus *Plasmodium*, with five species infecting humans (*P. falciparum*, *P. vivax*, *P. ovale*, *P. malariae*, and *P. knowlesi*), transmitted by female *Anopheles* mosquitoes [97]. *P. falciparum* is responsible for the most severe and often fatal cases, accounting for ~90% of the global burden, especially in sub-Saharan Africa [98].

In Europe, *P. vivax* was historically the predominant species, particularly in temperate areas. Extensive control measures, including marshland drainage and dichlorodiphenyltrichloroethane (DDT) use after World War II, eliminated endemic malaria, and the WHO declared the region malaria-free in 2015. Between 2000 and 2022, no malaria-related deaths were reported in Europe [99].

Globally, however, malaria remains a major public health challenge. According to WHO data, in 2023, an estimated 249 million malaria cases occurred across 85 endemic countries, marking an increase of 5 million cases compared to 2021. Although malaria cases had declined between 2000 and 2015, trends have reversed in recent years. The African Region remains the most affected, accounting for more than 95% of global cases, followed by Southeast Asia and the Eastern Mediterranean (2% each). The Americas and Western Pacific contribute the remaining burden [99].

Pediatric data on malaria are limited, particularly outside endemic regions. A study conducted in Côte d’Ivoire by Kouakou et al. assessed malaria distribution among children under five and pregnant women across different climatic zones. They found that malaria risk was significantly higher in humid tropical regions, with vulnerability influenced by environmental exposure and socio-demographic factors [100].

Key determinants of malaria transmission include the density and longevity of female *Anopheles* mosquitoes and their biting behavior (endophagic vs. exophagic) [101]. Adequate rainfall is necessary to form stable breeding sites, and ambient temperatures must support the parasite’s development within the vector. *P. vivax* requires temperatures of at least 15–16 °C, while *P. falciparum* requires 19–20 °C for sporogony within the mosquito to complete [102,103]. Thus, warm, humid environments with sufficient rainfall are optimal for both vector survival and parasite replication [104,105].

Rainfall, when moderate, provides breeding habitats for mosquitoes, but excessive precipitation can wash away larvae and reduce vector populations [106,107,108]. Similarly, temperature influences both the mosquito’s reproductive cycle and the sporogonic development of *Plasmodium* species within the vector [109].

The typical incubation period for malaria ranges from 7 to 21 days, though latency can be prolonged in semi-immune individuals or in infections by hypnozoite-forming species like *P. vivax* and *P. ovale*. A national reference center reported that 97% of imported malaria cases manifest within three months of return from endemic areas [110,111].

Initial symptoms include fever (in over 90% of cases), often accompanied by gastrointestinal (diarrhea, vomiting), neurological (headache, seizures), respiratory (cough), or renal (proteinuria) signs. Anemia, usually moderate, may also occur. Clinical findings are frequently nonspecific, with rare splenomegaly. Reassessment is important to detect progression to severe malaria or concomitant bacterial infections [110].

While the clinical picture of uncomplicated malaria is similar across *Plasmodium* species, some distinctions exist. *P. malariae* often causes milder symptoms, while *P. vivax* is linked to anemia and severe thrombocytopenia, particularly in children. Latency due to dormant liver-stage hypnozoites may lead to delayed symptom onset—up to four years in some *P. vivax* and *P. ovale* infections. *P. malariae*, although not forming hypnozoites, can persist in the bloodstream for years without symptoms before reactivation [110].

Congenital malaria is rare in non-endemic regions. In France, only 1 to 5 cases are reported annually. It typically results from maternal exposure to endemic areas and presents either as asymptomatic parasitemia or with signs like fever and jaundice in the first weeks of life [112].

Hyperreactive malarial splenomegaly (HMS), most often observed in children aged 2–5 years from unstable malaria regions, is characterized by chronic health decline, massive splenomegaly, moderate fever, and severe anemia, despite low or undetectable parasitemia. Polymerase chain reaction (PCR) or serologic tests are often needed for diagnosis [113].

Climate change is increasingly recognized as a major driver of malaria epidemiology. Studies using mathematical and geospatial models have projected shifts in transmission intensity and geographic distribution under future climate scenarios. Ayanlade et al. analyzed NOAA satellite data from 2000 to 2017 across six Nigerian ecological zones and identified precipitation as the most influential climatic factor. Malaria prevalence was higher in the wetter southern zones compared to drier northern regions. Non-climatic factors, such as irrigation, agriculture, migration, and urbanization, were also significant contributors to malaria transmission [114].

The Liverpool Malaria Model (LMM), developed by Hoshen and Morse in 2004, is a weather-driven simulation tool that predicts malaria transmission using daily mean temperatures and 10-day accumulated precipitation. It was later refined by Ermert et al. and applied to simulate malaria risk under various past and future climate conditions [115,116].

Diouf et al. applied the LMM to West Africa and demonstrated that rainfall and temperature variability drive seasonal malaria transmission peaks, with intense outbreaks linked to heavy rainfall events [117,118]. These models collectively indicate that by 2050, malaria distribution will shift toward higher altitudes and latitudes in response to warming climates and changing precipitation patterns.

Although climate change may increase the environmental suitability for malaria in regions like Europe, temperate Asia, and parts of North America, large-scale outbreaks are unlikely due to strong public health infrastructure, access to diagnostics and antimalarial therapies, and socio-economic resilience [119]. Nonetheless, malaria control programs must integrate climate modeling and surveillance to anticipate shifts in endemicity and adapt interventions accordingly.

### 4.5. Leishmaniasis

Leishmaniasis is a neglected tropical disease caused by protozoa of the genus *Leishmania* (family *Trypanosomatidae*), transmitted mainly by infected female sandflies—*Phlebotomus* spp. in the Old World and Lutzomyia spp. in the Americas [120]. Rare non-vector routes include transfusion, transplantation, and vertical transmission [120,121]. The disease occurs in three forms: cutaneous (CL), visceral (VL), and mucocutaneous (MCL), with distinct epidemiological profiles [120]. Risk is strongly linked to poverty, malnutrition, displacement, poor housing, and immunosuppression. Globally, 700,000–1 million new cases are reported annually, with CL and VL both endemic in parts of the WHO European Region and frequent imported cases [122].

In the EU, zoonotic CL and VL are usually due to *L. infantum* in the Mediterranean, while anthroponotic CL (*L. tropica*) occurs sporadically in Greece and nearby areas [123]. About 95% of CL cases arise in the Americas, Mediterranean, Middle East, and Central Asia, with ~70% concentrated in the Eastern Mediterranean, especially Iraq and Syria [124]. Conflict in Syria since 2013 has fueled major CL outbreaks and spread to neighboring countries, with refugee populations facilitating new transmission dynamics in Turkey, Lebanon, and Jordan [125,126,127,128].

Climate change has also been recognized as a major factor influencing the ecology and epidemiology of leishmaniasis. Rising temperatures and changes in humidity directly affect the development of *Leishmania* parasites within sandfly vectors, and indirectly impact the distribution and population density of the sandfly species themselves. For example, Ghatee et al. reported that the population density of *P. papatasi*, a vector of *L. major*, increases with arid conditions, suggesting a link between sandfly abundance and climate warming [125].

Temperature and humidity variations alter vector survival, reproductive cycles, and biting behavior. A study by Waitz et al. showed that sandfly population dynamics are strongly correlated with ambient temperatures, particularly with the mean temperature over the two weeks prior to collection, indicating a shortened lifecycle under higher temperatures [129,130]. For *P. papatasi*, both extremely high temperatures and very cold conditions limit activity; in addition, temperature modulates feeding frequency and vector competence [123,131].

Boussaa et al., in a study from Marrakech, Morocco, found that the optimal temperature range for *P. papatasi* activity lies between 32 and 36 °C, with peaks observed during the wetter months of the dry season (May and November) [132]. According to Ready, climate more strongly affects the distribution of cold-blooded sandfly vectors than the disease itself [133]. Nevertheless, climatic shifts, especially warming, have facilitated the expansion of sandflies to previously non-endemic areas, including higher altitudes such as the Atlas Mountains in Morocco, as observed by Guernaoui et al. [134].

In contrast, a cross-sectional study by Martin-Sánchez et al. in southern Spain (Alpujarras) suggested that temperature increases might influence transmission more than vector density, due to enhanced biting frequency and shortened parasite incubation periods within the vector [135,136]. These observations align with broader evidence indicating that the geographical distribution of *Phlebotomus* vectors in the Mediterranean is shifting in response to climate change [137].

Further evidence comes from a 2019 study by Erguler et al., who modeled sandfly population dynamics across Turkey, Cyprus, and Greece. Their findings underscored the significant role of land use changes and environmental modifications, such as irrigation and urbanization, in vector spread and habitat suitability [138].

Although the following descriptions are based on adult patients due to a lack of pediatric-specific data, it is known that the clinical manifestations of leishmaniasis vary with disease form and host characteristics. In CL, ulcerative skin lesions develop at the bite site weeks to months post-infection. While these may resolve spontaneously, they often leave disfiguring scars. In children, lesions may be more extensive and severe compared to adults [122].

VL is characterized by persistent fever, hepatosplenomegaly, anemia, leukopenia, and significant weight loss. Pediatric VL tends to present with more acute onset and severe clinical manifestations compared to adults, including rapid deterioration and higher risk of complications [139]. MCL, though rare, leads to destructive lesions in the nasal and oral mucosa, potentially resulting in serious deformities and respiratory obstruction. In children, these manifestations are less common but may be more difficult to manage when they occur [122].

The varied clinical spectrum highlights the importance of early diagnosis and individualized treatment based on disease form, patient age, and immune status. Preventive strategies remain centered on vector control and personal protection in endemic areas. Leishmaniasis continues to pose a major public health concern, exacerbated by environmental, climatic, and socio-political factors that influence its distribution and persistence across regions [140].

### 4.6. West Nile Virus

West Nile virus (WNV) is a climate-sensitive, multi-vector, multi-host arbovirus belonging to the family *Flaviviridae* [141,142]. It was first identified in 1937 in the West Nile district of Uganda, from which it derives its name [143]. The virus is primarily transmitted to humans through the bite of infected mosquitoes belonging to the genus Culex, especially *Culex pipiens*, *Cx. quinquefasciatus*, and *Cx. tarsalis* [142,144]. Birds, especially species such as egrets and herons, are the primary amplifying hosts, while humans and horses are considered incidental or “dead-end” hosts due to their low-level viremia, which is insufficient to sustain the transmission cycle [144,145].

WNV is part of the Japanese encephalitis serocomplex and is genetically classified into at least nine lineages. However, only lineages 1 and 2 are associated with human disease. Lineage 1 (L1) includes sublineage 1A, the most virulent strain for humans, which circulates in Europe, the Middle East, Africa, West Asia, and North America. Sublineage 1B is found in Oceania and is rarely neuroinvasive. Lineage 2 (L2) has historically circulated in sub-Saharan Africa and Madagascar but has been increasingly reported in Europe in recent decades [142].

WNV displays marked seasonality. The virus is amplified among bird populations during the spring and early summer and then spills over into humans during mid- to late summer, when mosquito densities peak [146]. *Culex pipiens*, the primary urban vector, reproduces in stagnant water sources enriched by organic matter. Drought conditions have been associated with increased transmission risk, as they concentrate organic material in breeding sites, reduce mosquito predators such as frogs and dragonflies, and increase bird aggregation around limited water resources, facilitating viral amplification [147].

In Europe, climatic anomalies have been identified as key drivers of WNV outbreaks and geographical expansion. In Italy, WNV was first identified in horses in 1998 and later in humans in 2008 [145]. Since then, southern European countries have reported a progressive increase in human cases, with surges corresponding to heatwaves and elevated temperatures [145,148]. In Romania, a major outbreak occurred in 1996, while other countries, such as Greece, Hungary, and Serbia, have reported increasing activity following the establishment of lineage 2 in the region. The 2018 season saw an unprecedented spike in cases, attributed to prolonged heat, ecological degradation from wildfires, and favorable environmental conditions [149,150].

The ongoing warming trend has extended the mosquito breeding season, reduced the extrinsic incubation period of the virus, and facilitated faster amplification among vectors and bird reservoirs [148]. Climate change has also influenced bird migration patterns, with a tendency for longer stops in more northerly regions, potentially introducing WNV into new territories [151]. A notable example is the 2020 detection of WNV in the Netherlands, a non-endemic country until then, highlighting the northward shift in the virus in Europe due to rising temperatures [5,141,142,148].

Clinically, approximately 80% of WNV infections are asymptomatic. In symptomatic cases, after an incubation period of 2 to 15 days, patients may present with nonspecific flu-like symptoms, including fever, headache, myalgia, nausea, and fatigue. Other features may include maculopapular or morbilliform rash, lymphadenopathy, conjunctival injection, vomiting, and, more rarely, orchitis [142]. In about 1 out of every 150 cases, the infection progresses to West Nile neuroinvasive disease (WNVND), which manifests as meningitis, encephalitis, or acute flaccid paralysis. These severe forms can result in long-term neurological sequelae or death, particularly among elderly, immunocompromised, or otherwise vulnerable individuals [142,152].

In children, the disease is usually mild and self-limiting. Nonetheless, some pediatric cases of WNVND with long-term sequelae or fatal outcomes have been documented, although children represent only about 5% of reported cases. The low rate could reflect underdiagnosis due to nonspecific symptoms or lower clinical suspicion [153,154]. Data in pediatric populations are mainly derived from surveillance in the United States, with fewer European studies available. Regardless of age, no specific antiviral treatment exists, and management remains supportive [155].

Preventive efforts focus on mosquito vector control and personal protection. While equine vaccines are available and widely used, no licensed human vaccine exists. Human vaccine candidates remain under investigation, with none beyond phase I or II trials [142]. Public health strategies rely on larviciding, the use of insecticides, reduction in mosquito breeding habitats, and environmental interventions. Simple measures such as eliminating standing water in containers and improving urban sanitation have been recommended. Notably, higher temperatures may favor viral evolution, promoting the emergence of immune-evading WNV strains, further stressing the importance of proactive surveillance and climate-sensitive vector control programs [144].

### 4.7. Other Arboviruses

Climate change is expected to influence the distribution and emergence of several lesser-known arboviruses of public health concern, including Usutu virus (USUV), Toscana virus (TOSV), and Sindbis virus (SINV), which are increasingly detected in Europe and neighboring regions [141].

Usutu virus, a *Flavivirus* closely related to WNV, is primarily transmitted by *Culex* mosquitoes and maintained in an enzootic cycle between birds and vectors. Similarly to WNV, humans are considered incidental hosts. While most infections are asymptomatic or mild, USUV has been associated with neuroinvasive disease, particularly in immunocompromised individuals [141]. Over the past two decades, USUV has spread across central and southern Europe, often detected in birds and mosquitoes before human cases are recognized, underscoring the importance of One Health–based surveillance.

Toscana virus, a *Phlebovirus* transmitted by *Phlebotomus* sandflies, is endemic in Mediterranean countries. It is one of the leading viral causes of meningitis and meningoencephalitis in southern Europe during summer, when vector activity peaks [141]. Despite its recognized burden, TOSV remains underdiagnosed due to limited awareness and diagnostic availability outside endemic regions. Climate-driven expansions of sandfly habitats may increase the incidence of TOSV infections in temperate Europe.

Sindbis virus, an *Alphavirus* transmitted by *Culex* mosquitoes, circulates in enzootic cycles involving birds and has caused human outbreaks in northern Europe, particularly in Finland and Sweden. SINV infection, also known as “Pogosta disease,” typically manifests as febrile illness with rash and arthralgia, sometimes persisting for months [142]. Although historically restricted to northern latitudes, warmer conditions and shifts in bird migration routes could facilitate wider geographic spread.

Taken together, these arboviruses illustrate how climate change, by altering vector distribution, migration of avian reservoirs, and seasonal activity, may increase the risk of spillover into human populations. Although currently less widespread than DENV or WNV, their growing detection in Europe signals an urgent need for strengthened surveillance, improved diagnostics, and integration into public health preparedness plans. Early recognition will be essential to prevent underestimation of their true burden in a changing climate [141,142].

## 5. Pediatric Considerations and Data Gaps

Children represent a particularly vulnerable group in the context of climate-sensitive VBD. Across the infections discussed in this review, pediatric patients often show distinctive clinical manifestations and may experience more severe complications compared to adults. For dengue, children frequently present with nonspecific symptoms such as irritability, diarrhea, and convulsions, alongside higher risks of plasma leakage and severe disease progression [48,49,50,51]. Chikungunya in children may include prolonged joint symptoms, febrile seizures, and, in the case of congenital or perinatal infections, severe systemic complications [73,74]. Zika virus poses unique risks to children through congenital Zika syndrome, with long-term neurodevelopmental impairments beyond structural birth defects [84,89]. Malaria in children is associated with rapid progression to severe anemia, splenomegaly, and neurological complications [100,110,111,112,113], while pediatric visceral leishmaniasis tends to present more acutely and with higher complication rates than in adults [122,139]. Although WNV is usually mild in children, rare cases of neuroinvasive disease with long-term sequelae have been documented [153,154].

Despite these observations, substantial data gaps persist regarding the pediatric burden of many arboviral and parasitic infections in Europe. Most available evidence comes from adult populations or small pediatric cohorts, limiting the ability to accurately estimate incidence, severity, and outcomes in children [5,6,141]. This underlines the importance of strengthened surveillance systems that systematically disaggregate data by age group and integrate child-specific outcomes. Enhanced pediatric surveillance would not only clarify disease impact but also serve as an early warning indicator of shifting epidemiological patterns under climate change.

It is also important to emphasize that climate change may disproportionately affect children beyond pathogen-specific risks. Increased outdoor exposure, limited use of protective measures, immature immune responses, and higher vulnerability to dehydration during febrile illness place children at elevated risk [48,73,84]. Social and environmental factors—including displacement, poor housing, and limited healthcare access—further amplify these vulnerabilities, particularly in the aftermath of heatwaves, floods, or other climate-related events [84,125,126,127,128].

In this context, the focus on pediatric implications represents an epidemiologically valuable contribution. By integrating child-centered perspectives into surveillance, predictive models, and public health planning, the scientific and medical community can better anticipate emerging threats and design interventions that protect the youngest and most vulnerable in a changing climate.

## 6. Conclusions

Climate change is increasingly recognized as one of the most pressing global threats, with profound implications not only for environmental stability but also for human health. The acceleration of global warming observed in recent decades has created ecological conditions that are highly conducive to the emergence, re-emergence, and spread of infectious diseases, particularly those transmitted by arthropod vectors. Arboviruses and other VBDs have shown a concerning tendency to expand into previously non-endemic areas, driven by rising temperatures, altered precipitation patterns, and the increased movement of vectors and hosts.

Climatic factors significantly modulate the biology, survival, and distribution of vectors, as well as the replication and transmission efficiency of the pathogens they carry. Temperature fluctuations, for instance, influence the biting behavior and reproductive cycles of vectors, reduce the extrinsic incubation period of viruses, and alter the overall vectorial capacity, thereby increasing the risk of sustained local transmission. In Europe, including Italy, climate-induced shifts in environmental parameters have been associated with a growing number of autochthonous cases of diseases such as dengue, chikungunya, West Nile virus, leishmaniasis, and malaria, underscoring the urgent need for tailored public health responses.

To mitigate these risks and anticipate future challenges, further research is essential. High-resolution predictive models that integrate climate, ecological, and epidemiological data are needed to improve forecasting and early warning systems. Longitudinal studies should investigate how climate change influences vector competence and pathogen evolution, particularly in relation to changes in extrinsic incubation periods and geographic distribution. Expanding epidemiological data for pediatric populations, currently underrepresented in many studies, is also a priority, especially for emerging arboviruses. Furthermore, evaluating the effectiveness of vector control strategies in diverse environmental contexts and understanding socio-environmental determinants such as urbanization, land use, and migration patterns will enhance targeted interventions. Public health preparedness must also be strengthened by incorporating climate projections into national health policies and disease prevention frameworks. Finally, continuous genomic surveillance of emerging viruses will be critical to detect new variants, monitor viral evolution, and assess the potential for vaccine and treatment resistance.

In light of these considerations, the interconnection between climate change and emerging infections demands a multidisciplinary, collaborative, and globally coordinated approach. It is not only a scientific and medical challenge but also a societal and ethical imperative to ensure the protection of human health—particularly for vulnerable populations—in an increasingly unstable climate. Only through sustained investment in research, surveillance, and evidence-based public health action can we hope to meet this challenge and secure a safer future for the generations to come.

## Figures and Tables

**Table 1 microorganisms-13-02034-t001:** Climate Change Impact on Vector-Borne Diseases.

Disease	Temperature Increase (°C)	Vector Range Expansion	Incubation Period Change	Impact on Transmission Rate
West Nile Virus	2–3 °C	Yes	Decreased	Higher
Malaria	2–4 °C	Yes	Decreased	Higher
Dengue	2–3 °C	Yes	Decreased	Higher
Chikungunya	2–3 °C	Yes	Decreased	Higher
Leishmaniasis	1–2 °C	Yes	Decreased	Higher
Zika	2–3°	Yes	Decreased	Higher

**Table 2 microorganisms-13-02034-t002:** Emerging Vector-Borne Diseases (VBDs) in Europe.

Disease	Vector	Host	Regions of Spread	New Areas of Spread	Climate Factors Influencing Spread
Dengue	*Aedes* mosquitoes	Humans	Southeast Asia, Latin America, Africa	Southern Europe (e.g., Spain, Italy)	Temperature increase, increased rainfall, urbanization, mosquito breeding sites
Chikungunya	*Aedes* mosquitoes	Humans	Africa, Asia, Americas	Caribbean, Mediterranean regions	Warmer temperatures, changes in rainfall patterns, urbanization
Zika Virus	*Aedes* mosquitoes	Humans	Americas, Southeast Asia	U.S. territories (e.g., Puerto Rico)	Temperature increase, altered habitats, urbanization
Malaria	*Anopheles* mosquitoes	Humans	Sub-Saharan Africa, Southeast Asia	Southern Europe (e.g., Greece)	Increased rainfall, warmer temperatures, changes in land use
Leishmaniasis	Sandflies	Humans, dogs	Mediterranean region, parts of Asia and Africa	Southern Europe, urban areas in the Middle East	Rising temperatures, habitat changes, urbanization
West Nile Virus	*Culex* mosquitoes	Birds, humans	North America, Europe, Africa	Expanded into parts of Europe	Temperature increase, altered precipitation patterns, habitat availability

## Data Availability

No new data were created or analyzed in this study.

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
