# Peer review of "Vectors on the Move: How Climate Change Fuels the Spread of Arboviruses in Europe"

_microorganisms, 2025, doi:10.3390/microorganisms13092034_

Round 1

Reviewer 1 Report

Comments and Suggestions for Authors

Table formatting needs to be improved. Species need to be in italics—this is throughout the entire paper--. Line 178 EIP has already been defined

Formatting Line 224

226 Genus Alphavirus needs to be italized

Line 239: abbreviate the genus same for 242 and italics

Line 255 EIP has already been defined

Line 259 VBD has already been defined

Line 311 abbreviate the genus

Line 324 replace incubation with EIP

Line 603 formatting

Line 615 use EIP

The authors could strengthen pediatric focus by adding a dedicated subsection summarizing all pediatric findings and data gaps.  Furthermore, the authors could discuss how climate change may disproportionately affect children’s risk, exposure, and disease severity beyond what is presented for each pathogen.

The literature search strategy is described, but no PRISMA-style flow diagram or clear count of initial vs. included studies is presented.  The authors could include a flow diagram and brief note on potential selection bias from database/language restrictions.

While the narrative format allows flexibility, some pathogen sections (e.g., dengue, chikungunya) are far more detailed than others (e.g., leishmaniasis, Zika).  The authors could balance depth by either condensing the longest sections or expanding the shorter ones, ensuring each covers:

    1. European epidemiology and climate influence
    2. Pediatric-specific features
    3. Surveillance/control strategies

Author Response

Table formatting needs to be improved. Species need to be in italics—this is throughout the entire paper--. Line 178 EIP has already been defined
Re: Revised as recommended.

Formatting Line 224: 
Re: Done.

226 Genus Alphavirus needs to be italized.
Re: Done.

Line 239: abbreviate the genus same for 242 and italics
Re: Done.

Line 255 EIP has already been defined
Re: Revised.

Line 259 VBD has already been defined
Re: Revised.

Line 311 abbreviate the genus
Re: Done.

Line 324 replace incubation with EIP
Re: Revised.

Line 603 formatting
Re. Revised. 

Line 615 use EIP
Re: Revised.

The authors could strengthen pediatric focus by adding a dedicated subsection summarizing all pediatric findings and data gaps.  Furthermore, the authors could discuss how climate change may disproportionately affect children’s risk, exposure, and disease severity beyond what is presented for each pathogen.
Re: A pediatric subsection has been added as recommended (pp. 15-16).
The literature search strategy is described, but no PRISMA-style flow diagram or clear count of initial vs. included studies is presented.  The authors could include a flow diagram and brief note on potential selection bias from database/language restrictions.
Re: This is a narrative review, not a systematic one. However, further details has been added in the Methods (pp. 2-3).

While the narrative format allows flexibility, some pathogen sections (e.g., dengue, chikungunya) are far more detailed than others (e.g., leishmaniasis, Zika).  The authors could balance depth by either condensing the longest sections or expanding the shorter ones, ensuring each covers:
1.    European epidemiology and climate influence
2.    Pediatric-specific features
3.    Surveillance/control strategies

Re: Revised as recommended (pp. 5-15).

Reviewer 2 Report

Comments and Suggestions for Authors

Type of manuscript: Review

Title: Vectors on the Move: How Climate Change Fuels the Spread of Arboviruses in Europe

Journal: Microorganisms

Authors: Giulia Carbone, Giulia Boiardi, Claudia Infantino, Daniela Cunico, Susanna Esposito

The review presented in this manuscript on the most epidemiologically relevant vector-borne diseases (VBDs) is very interesting. The topic is highly relevant, especially for European Union countries that are currently experiencing outbreaks of autochthonous cases of arboviruses. It is important to emphasize the need for strengthened surveillance of these diseases in the context of climate change. The focus on pediatric implications also represents an epidemiologically valuable contribution.

However, I would suggest that the authors revise the organization of the paragraphs describing every VBDs. This would avoid unnecessary repetitions and allow common aspects of VBDs to be addressed more effectively.

Attention should also be paid to the use of acronyms (for example, EIP is sometimes spelled out and sometimes not; it would be better to standardize), as well as to the explanation of acronyms when they first appear (e.g., DHF/DSS, lines 203, 210). Scientific names should consistently be written in italics at the genus and species level, but not at the family level.

Since the review lacks a discussion on tick-borne diseases, it would be advisable to provide an explanation for this choice. The Methods section indicates that the search also included ticks (line 67), but not the diseases they transmit (line 68). There are only brief mentions of ticks in lines 97 and 133, and of Lyme disease in line 134.

A thorough revision of the reference list is necessary, as there are several inconsistencies and typographical errors. Some references lack authors (e.g., reference n. 9, line 674 and onwards); in others, only the authors’ initials are provided (e.g., reference n. 1, line 656 and onwards). Genus and species names are often not italicized, and several typos are present (e.g., reference n. 101, line 893; reference n. 102, line 895).

Below, I provide some detailed observations:

  • Lines 47–50: please rephrase the following passage: “Warming temperatures and altered rainfall have facilitated the northward expansion of vectors such as Aedes spp. mosquitoes and Phlebotomus spp. sandflies [6]. This has increased the risk of local transmission of diseases previously considered exotic, including dengue, chikungunya, Zika, leishmaniasis, and West Nile virus.”
    West Nile virus is transmitted by Culex mosquitoes, not Aedes.  Leishmaniasis and West Nile virus are already autochthonous VBDs in Europe.
  • Line 154: “genus” should not be italicized.
  • Line 372: correct the year with 2023 “According to WHO data, in 2023 an estimated…”.
  • Line 396: please specify which: “A national reference center…”.
  • Line 523: separate the two sections: 4.6 West Nile Virus and 4.5 Other arboviruses.
  • From line 589 onwards: this section requires clearer explanation. For example, Usutu virus, like Sindbis virus, is transmitted by Culex

Author Response

The review presented in this manuscript on the most epidemiologically relevant vector-borne diseases (VBDs) is very interesting. The topic is highly relevant, especially for European Union countries that are currently experiencing outbreaks of autochthonous cases of arboviruses. It is important to emphasize the need for strengthened surveillance of these diseases in the context of climate change. The focus on pediatric implications also represents an epidemiologically valuable contribution.
Re: Thank you very much for your positive evaluation. Also considering the other reviewer’s comments, we further strengthened the pediatric considerations (pp. 15-16).

However, I would suggest that the authors revise the organization of the paragraphs describing every VBDs. This would avoid unnecessary repetitions and allow common aspects of VBDs to be addressed more effectively.
Re: We improved the text maintaining a balance in the description of each VBD, considering European epidemiology and climate influence, pediatric-specific features, and surveillance/control strategies (pp. 5-15). 

Attention should also be paid to the use of acronyms (for example, EIP is sometimes spelled out and sometimes not; it would be better to standardize), as well as to the explanation of acronyms when they first appear (e.g., DHF/DSS, lines 203, 210). Scientific names should consistently be written in italics at the genus and species level, but not at the family level.
Re: We revised the text throughout the manuscript as suggested.

Since the review lacks a discussion on tick-borne diseases, it would be advisable to provide an explanation for this choice. The Methods section indicates that the search also included ticks (line 67), but not the diseases they transmit (line 68). There are only brief mentions of ticks in lines 97 and 133, and of Lyme disease in line 134.
Re: Clarified in the Methods (p. 2).

A thorough revision of the reference list is necessary, as there are several inconsistencies and typographical errors. Some references lack authors (e.g., reference n. 9, line 674 and onwards); in others, only the authors’ initials are provided (e.g., reference n. 1, line 656 and onwards). Genus and species names are often not italicized, and several typos are present (e.g., reference n. 101, line 893; reference n. 102, line 895).
Re: You are absolutely right. We revised the references as recommended.

Below, I provide some detailed observations:
•    Lines 47–50: please rephrase the following passage: “Warming temperatures and altered rainfall have facilitated the northward expansion of vectors such as Aedes spp. mosquitoes and Phlebotomus spp. sandflies [6]. This has increased the risk of local transmission of diseases previously considered exotic, including dengue, chikungunya, Zika, leishmaniasis, and West Nile virus.”
West Nile virus is transmitted by Culex mosquitoes, not Aedes.  Leishmaniasis and West Nile virus are already autochthonous VBDs in Europe. Re: Revised.
•    Line 154: “genus” should not be italicized. Re: Revised.
•    Line 372: correct the year with 2023 “According to WHO data, in 2023 an estimated…”. Re: Revised.
•    Line 396: please specify which: “A national reference center…”. Re: Specified.
•    Line 523: separate the two sections: 4.6 West Nile Virus and 4.5 Other arboviruses. Re: Revised as suggested.
•    From line 589 onwards: this section requires clearer explanation. For example, Usutu virus, like Sindbis virus, is transmitted by Culex Re: Clarified as recommended.